# *BiomedParse-V*: Scaling Foundation Model for Universal Text-guided Volumetric Biomedical Image Segmentation

Theodore Zhao[1], Ho Hin Lee[1], Alberto Santamaria-Pang[1,2], Noel C. Codella[1], Sid Kiblawi[1], Yu Gu[1], Yu Fang[1], Wen Xuan Teng[1], Naiteek Sangani[1], Ivan Tarapov[1], Matthew P. Lungren[1], Matthias Blondeel[1], Tristan Naumann[1], Naoto Usuyama[1], Sheng Wang[1,3], Paul Vozila[1], Hoifung Poon[1], and Mu Wei[1]

[1] Microsoft, Redmond WA 98052, USA
[2] Johns Hopkins Medicine, Baltimore MD 21205, USA
[3] University of Washington, Seattle WA 98195, USA
MuHsin.Wei@microsoft.com
Code available at https://github.com/microsoft/BiomedParse

**Abstract.** Three-dimensional (3D) image segmentation plays a pivotal role in clinical diagnosis, therapy planning, and drug discovery by enabling the precise delineation of anatomical structures, pathological lesions, and cellular features in medical imaging modalities such as CT and MRI, as well as in biomedical microscopy. Despite its central importance, 3D segmentation remains a formidable technical challenge due to high computational requirements, the vast diversity of segmentation tasks across clinical and research domains, and the lack of interoperability among existing models, which are typically developed for specific modalities and tasks. To address these limitations, we introduce *BiomedParse-V*, a scalable and generalizable multimodal foundation model that leverages pretrained 2D foundation models to enable accurate, text-prompted 3D image segmentation. Our method features a novel Fractal Volumetric Encoding (FVE) scheme, which hierarchically compresses volumetric data by capturing self-similarity across slices into a compact fractal-based 2.5D representation. This design allows the effective use of powerful 2D foundation models while preserving essential 3D spatial context. We further propose the Independent Segmentation Discriminator (ISD) module to promote robust and consistent object localization throughout the segmented volume, addressing the challenges of maintaining spatial coherence in text-guided segmentation. Extensive experiments conducted across diverse biomedical imaging modalities demonstrate that *BiomedParse-V* consistently achieves state-of-the-art performance, significantly surpassing leading supervised 3D segmentation models. Our approach delivers a prompt-driven, computationally efficient, and broadly applicable solution for 3D biomedical image segmentation, advancing the accessibility and impact of segmentation technologies in real-world clinical and research environments.

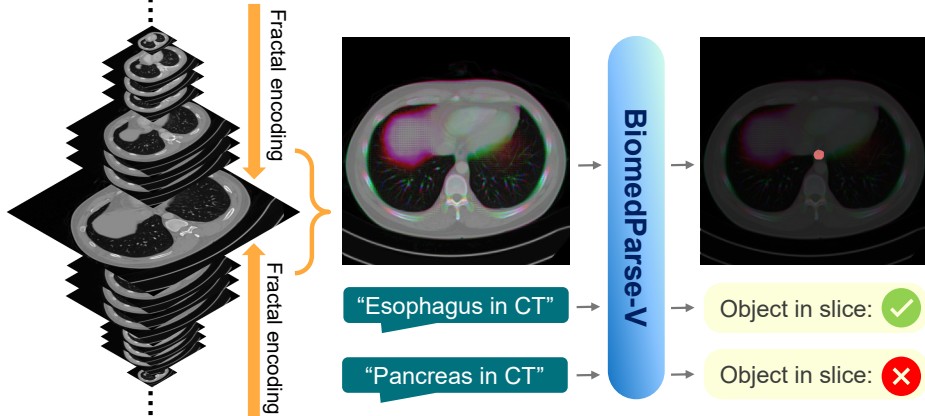

**Fig. 1.** Overview of *BiomedParse-V*. Volumetric medical data is encoded using Fractal Volumetric Encoding, compressing rich 3D context into an RGB format. Text-prompted segmentation is performed on the encoded slices, and the presence of the prompted object within each slice is predicted by the Independent Segmentation Discriminator.

## 1   Introduction

Biomedical image segmentation is the cornerstone of both clinical medicine and biomedical research, allowing accurate diagnosis, therapy planning, and disease monitoring. In radiology, three-dimensional (3D) segmentation of modalities such as Computed Tomography (CT) and Magnetic Resonance Imaging (MRI) provides detailed anatomical and pathological insights, while in microscopy, 3D segmentation is critical for cellular and subcellular analysis. However, the diversity of imaging modalities and segmentation targets introduces substantial technical challenges. Existing models are typically developed for specific modalities, resulting in limited generalization and poor interoperability across domains, a significant barrier in multidisciplinary settings where seamless integration of radiology and microscopy data is essential.

Recent advances in foundation models have shown promise for unified, prompt-driven segmentation frameworks that use natural language guidance to address multiple tasks. Methods such as MedSAM [15] leverage large-scale pre-trained models to generalize across a variety of segmentation scenariosthrough interactive prompting. However, adapting these models to 3D volumetric data is hampered by increasing interaction demands from the users. Furthermore, promptable 3D segmentation approaches such as SegVol [3] often underperform when using textual prompts alone, due to difficulties in contextually aligning language with volumetric image data. Addressing these challenges is crucial for developing scalable and interoperable segmentation solutions applicable across the full spectrum of biomedical imaging modalities. To address these challenges,

we argue that traditional volumetric processing at a fixed spatial scale is inefficient, necessitating carefully designed network architectures that implicitly learn transformations equivalent to data encodings. We challenge the assumption that volumetric information cannot be utilized by 2D model architectures. We drew inspiration from the mathematical concept of fractal, where infinity can be encoded in a finite space, such as the Sierpiński carpet [22] and the Koch curve [23], defying the traditional notion of dimensionality. Inspired by this insight, we introduce *Fractal Volumetric Encoding (FVE)*, a novel encoding strategy that recursively compresses 3D volumes into patterns of reducing scale on the 2D plane and enables leveraging pretrained 2D foundation models.

Another major challenge in text-prompted 3D segmentation is accurately determining whether the prompted anatomical structures are present in each individual slice, as errors can result in numerous false-positive predictions. To address this, we propose an *Independent Segmentation Discriminator (ISD)* module, which independently verifies the presence of the prompted structures per slice basis. By effectively filtering irrelevant slices, the ISD significantly reduces false positives and enhances the overall reliability of text-prompted segmentation. Our key contributions are summarized as follows:

– We propose **Fractal Volumetric Encoding (FVE)**, a novel method that efficiently encodes the 3D volumetric context into a compact 2.5D representation, allowing the use of powerful and efficient 2D foundation models for volumetric segmentation while preserving rich spatial information.
– We introduce the Independent Segmentation Discriminator (ISD) module to robustly determine slice-level segmentation relevance, enhancing the accuracy of text-promptable segmentation in 3D volumes.
– We present *BiomedParse-V*, a single multimodal segmentation foundation model that brings the power of BiomedParse to volumetric segmentation using FVE and ISD, and show through extensive validation of various 3D modalities that *BiomedParse-V* achieves superior segmentation accuracy compared to current state-of-the-art 3D segmentation methods.

## 2   Related work

Medical image segmentation has continually advanced to improve accuracy across diverse imaging modalities and clinical challenges. Early advances, such as the introduction of U-Net and its 3D variants [21,1], established a robust foundation for supervised dense segmentation. These led to specialized architectures, including UNet++ [31], H-DenseUNet [12], and SegResNet [19], optimized for specific anatomical and modality-driven contexts. More recently, frameworks such as nnU-Net [7] introduced comprehensive hierarchical designs to improve coarse-to-fine learning.

Alongside these developments, transformer-based architectures have gained prominence driven by their capacity for modeling long-range dependencies. Models such as UNETR [5] leverage vision transformers (ViTs) to capture broad spatial relationships across medical images, at the cost of significant computational

overhead. To mitigate this, hierarchical transformer designs such as the Swin Transformer, used in SwinUNETR [4], employ efficient sliding-window strategies for localized attention. Depthwise convolution-based architectures, such as 3D UX-Net [11], offer an alternative path to computational efficiency, nevertheless their performance remains limited when addressing anatomical structures that vary across multiple spatial scales. Foundation models have recently transformed medical image segmentation, enabling flexible, task-agnostic segmentation guided by diverse prompts, including natural language. Notable examples include MedSAM [15] and BiomedParse [28], which build upon large-scale pretrained general-domain models to achieve remarkable segmentation performance. However, extending these prompt-driven models to 3D volumetric imaging remains challenging due to significantly increased computational and memory demands.

Prompt-based segmentation frameworks specifically targeting 3D modalities, such as SegVol [3], have employed spatial prompts such as bounding boxes and points to guide the model and improve segmentation accuracy. However, their effectiveness decreases when relying solely on textual prompts. This limitation arises due to inherent challenges in aligning linguistic semantics with complex 3D volumetric structures, revealing a critical gap for text-based prompting methods in volumetric contexts. Effective data encoding strategies are crucial to improve segmentation efficiency and accuracy. Recent advances in representation learning, particularly transformer-based architectures and self-supervised methods, have greatly enriched feature expressiveness and quality. Despite these advances, explicit and efficient encoding of volumetric medical data within prompt-driven segmentation frameworks remains largely unexplored, representing an important research frontier.

In general, current segmentation methods exhibit critical limitations: specialized modality-specific models lack interoperability, computational complexity restricts practical deployment, and existing prompt-based methods struggle to contextually align textual prompts in 3D data. Addressing these gaps requires novel approaches capable of efficiently compressing volumetric information, enhancing contextual semantic alignment, and promoting interoperability across imaging modalities. In contrast, *BiomedParse-V* directly addresses these challenges by introducing fractal-informed representations, efficiently compressing volumetric data, and allowing robust text-based prompt segmentation in a unified framework.

## 3   Method

Our proposed *BiomedParse-V* framework addresses the computational and scalability challenges inherent in volumetric biomedical image segmentation by leveraging efficient fractal-informed encoding and robust prompt-driven segmentation. The general architecture is illustrated in Fig. 1. Specifically, using FVE, a focal slice is encoded along with its corresponding 3D context into a compact RGB representation, enabling effective segmentation guided by text prompts

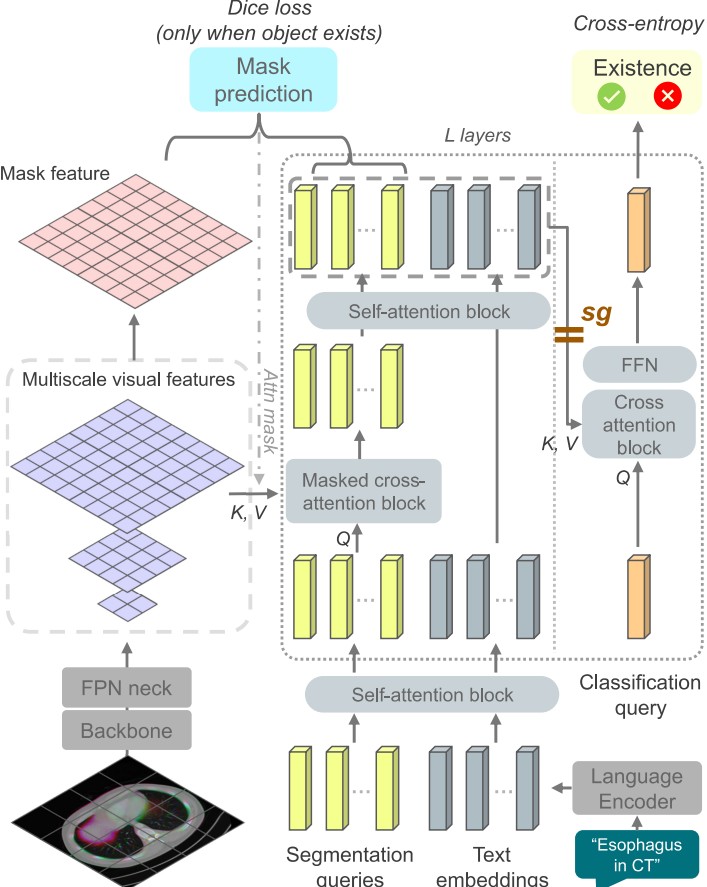

**Fig. 2.** The model architecture of *BiomedParse-V*. For each slice in the 3D volume, we encode spatial context around the slice into an RGB image (bottom left) and feed it into the model. The model then performs segmentation on the slice based on the given text prompt (bottom right). A transformer mask decoder processes the image features and text embeddings along with learnable segmentation queries and a classification query to produce the mask prediction and verify the existence of the prompted object.

without additional computational overhead. To further improve segmentation reliability, we predict the presence of the prompted object within encoded slices using the dedicated ISD classification module.

### 3.1   Promptable Segmentation Framework

Given a 3D image volume $\mathcal{V} \in \mathbb{R}^{H \times W \times D}$ and text prompt $\mathcal{P}$ describing the desired object, the promptable segmentation problem is to output a binary mask

$$M = f(\mathcal{V}, \mathcal{P}) \in \{0, 1\}^{H \times W \times D},$$

where the set of 1's corresponds to the voxels occupied by the described object.

We leverage 2D foundation model architectures to perform slice-by-slice segmentation, while encoding volumetric context into a single image. Following recent advances in segmentation foundation models [10,32,15,27], we adopt a transformer-based decoder architecture. As depicted in Fig. 2, we utilize a vision backbone to extract multi-scale visual features further refined by a Feature Pyramid Network (FPN) [13]. Text prompts are transformed into text embeddings and combined with learnable segmentation queries, which are processed iteratively through multiple transformer blocks. In each transformer layer, segmentation queries cross-attend to visual features at different scales. Specifically, we adopt the Boltzmann attention sampling strategy from BoltzFormer [29] to maintain segmentation quality even when the object appears small in the slice.

Additionally, a dedicated classification query in parallel to the transformer decoder independently assesses object existence, reducing false positives in text prompt-driven segmentation tasks, which is discussed in detail in Section 3.4.

In our experiments, we used FocalNet [25] as the image backbone which outputs four scales of features with strides 4, 8, 16 and 32. The FPN takes the multiscale backbone features to output multiscale visual features of strides 4, 8, 16 and 32. The multiscale visual features convolute to mask feature of stride 4. The segmentation transformer block attends to multiscale visual features of strides 16, 8, and 4 in 3 loops, totaling up to nine layers. We interpolated to original resolution of the image shape.

### 3.2   Fractal Volumetric Encoding

To efficiently embed volumetric medical data into a compact representation suitable for 2D foundation models, we propose fractal volumetric encoding. Formally, this fractal encoding process is expressed as

$$\mathcal{E}(\mathcal{V}, i) : \mathbb{R}^{H \times W \times D} \otimes \mathbb{N} \to \mathbb{R}^{H \times W \times C},$$

where $\mathcal{E}$ transforms the 3D voxel data $\mathcal{V}$ into a multi-channel 2D image, centered around the $i$-th slice. Although this encoding scheme is general and supports arbitrary channel dimensions, we specifically focus on encoding into RGB channels ($C = 3$) so to leverage and be compatible with common 2D vision architectures.

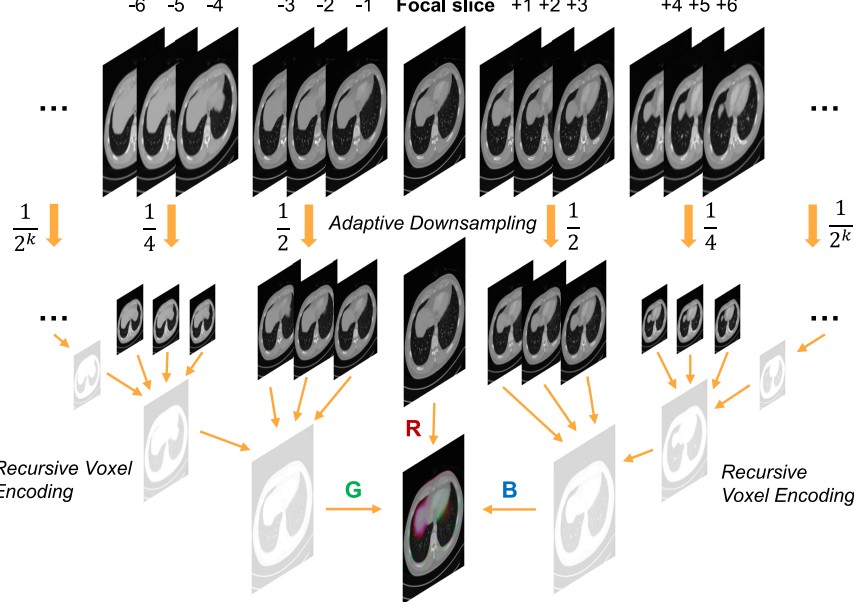

**Fig. 3.** Illustration of the fractal volumetric encoding process. A focal slice is combined with its surrounding 3D context. The focal slice maintains full resolution, while the context is adaptively downsampled and encoded recursively into compact channels, ensuring efficient spatial representation.

Specifically, the focal slice $\mathcal{V}_i$ (target segmentation plane) is stored in one RGB channel at full resolution, while adjacent upper and lower context slices are compressed into the other two channels via fractal-based hierarchical compression. This recursive encoding partitions slices into hierarchical groups, applying adaptive downsampling proportional to their distance from the focal slice. The process, depicted in Fig. 3, efficiently preserves critical spatial coherence with a limited channel bandwidth. The pseudo code to encode one side of the 3D context into a single channel is described in Algorithm 1. The fractal encoding order $k$ determines the number of slices used.

---

**Algorithm 1** Fractal Volumetric Encoding

---
1: **Input:** A list of image slices imgs $= [s_1, s_2, \cdots]$, ordered by their distance to the focal slice. Fractal encoding order $k$.
2: **Output:** A single image of the same resolution as the slices.
3: **def** fractal_encode(imgs, $k$):
4:    **if** $k > 1$:
5:        imgs_low = Downsample imgs by half
6:        $\mu_1, \mu_2, \mu_3 =$ imgs_low[:3]
7:        $\mu_4 =$ fractal_encode(imgs_low[3:], $k - 1$)
8:        **return** pixel_mix($\mu_1, \mu_2, \mu_3, \mu_4$)
9:    **else:**
10:        **return** imgs[0]

---

To construct a fractal-encoded image, we define the pixel_mix function that combines 4 images into a single image with twice the resolution:

$$\mu' = \text{pixel\_mix}(\mu_1, \mu_2, \mu_3, \mu_4),$$

where $\mu_1, \mu_2, \mu_3, \mu_4 \in \mathbb{R}^{h \times w}$ and $\mu' \in \mathbb{R}^{2h \times 2w}$. The pixel-level mapping is as follows:

$$\mu'(2x, 2y) = \mu_1(x, y)$$
$$\mu'(2x, 2y + 1) = \mu_2(x, y)$$
$$\mu'(2x + 1, 2y) = \mu_3(x, y)$$
$$\mu'(2x + 1, 2y + 1) = \mu_4(x, y),$$

for $x = 0, \cdots, h - 1$ and $y = 0, \cdots, w - 1$.

For an encoding of order $k$, the total number of slices encoded per side is $N(k) = 3k - 2$. Including the focal slice, the RGB-encoded image represents a total of $N_{rgb}(k) = 6k - 3$ slices. Padding is applied by repeating the last available slice if needed.

### 3.3   Mathematical Equivalence to 3D Convolution

To theoretically justify the proposed fractal encoding, we establish its mathematical equivalence to traditional 3D convolution operations commonly employed in volumetric segmentation methods.

Patch-based vision models, such as ViT [2] and FocalNet[25], typically process an input image by dividing it into disjoint patches (resolution $p \times p$). Each patch $P_{ij} \in \mathbb{R}^{p \times p \times C}$ is embedded with a linear projection as follows:

$$e_{ij} = \text{Linear}\left(\text{Flatten}(P_{ij})\right),$$

where $(i, j)$ denotes the spatial indices of the patch. The network then processes the patch embeddings $e_{ij}$ in units.

Within our fractal encoding, each 2D patch aggregates voxels aligned along the depth dimension ($z$-axis), forming compact neighborhoods we refer to as "super pixels". Mathematically, each patch $P$ captures a voxel neighborhood $V$, as illustrated in Fig. 4.

We formally establish the mathematical equivalence between applying linear transformations (patch embeddings) on fractal-encoded patches and performing an adaptive-resolution 3D convolution operation.

**Theorem 1 (Equivalence to 3D Convolution).** *Given a fractal encoding function $\mathcal{E}$, defined as*

$$\mathcal{E}(\mathcal{V}, i) : \mathbb{R}^{H \times W \times D} \otimes \mathbb{N} \to \mathbb{R}^{H \times W \times C}, \tag{1}$$

*applying a linear embedding operator on patches of the encoded image corresponds mathematically to performing an adaptive-resolution 3D convolution on the original volume.*

*Proof.* We leave the formal proof to the appendix.

To ensure equivalence, each "super pixel" must be fully contained within a single model patch. Consequently, the minimum patch size is determined by the encoding order $k$ and is given by $2^{k-1}$.

This analysis demonstrates that fractal encoding inherently represents an adaptive, multi-scale 3D convolution operation, efficiently rearranged into a patch-compatible multi-channel (e.g., RGB) format suitable for standard 2D vision architectures.

### 3.4   Independent Segmentation Discriminator (ISD)

A major challenge for purely text-prompted segmentation in 3D is to reliably determine the presence of prompted objects within each slice. To overcome this, we propose the Independent Segmentation Discriminator (ISD), which uses a dedicated classification query vector $q$ in parallel to the segmentation transformer decoder, as shown on the right side in Fig. 2. $q$ cross-attends to the hidden state of the segmentation queries and text embeddings at the end of each transformer block and then updates itself through a feed-forward network (FFN).

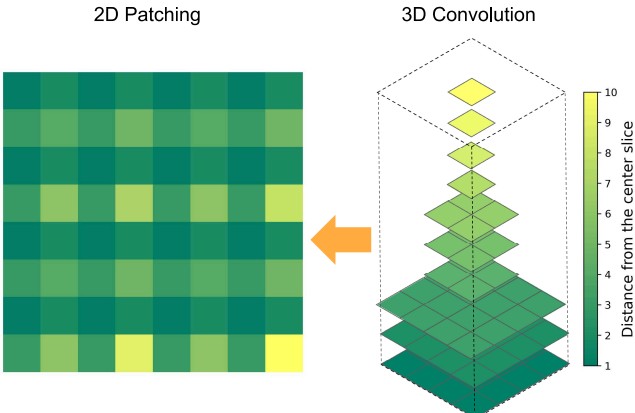

**Fig. 4.** The 2D patching process on fractal-encoded channels is mathematically equivalent to adaptive-resolution 3D convolution. Each colored square represents a pixel/voxel within a 2D patch or corresponding 3D voxel neighborhood. Pixel colors indicate distance from the focal slice. This visualization depicts the encoding of one side of the 3D context.

After the last transformer block, the $q$ produces a binary classification probability $\omega \in [0, 1]$ through a linear layer, representing the existence of the prompted object. At inference time, the final segmentation mask prediction $\hat{M}$ for a given slice is computed as:

$$\hat{M} = \hat{M}_{raw} \cdot (\omega > 0.5). \tag{2}$$

The ISD is trained jointly with the segmentation module to classify the existence of the prompted object based on the transformer's hidden states during segmentation. $\omega$ is supervised jointly with the segmentation prediction $\hat{M}_{raw}$ through the following loss function:

$$\mathcal{L}(\hat{M}_{raw}, \omega; M) = \begin{cases} \mathcal{S}(\hat{M}_{raw}, M) - \log(\omega), & M \neq 0 \\ -\log(1 - \omega), & M = 0. \end{cases} \tag{3}$$

Since segmentation masks are supervised only on positive slices using segmentation loss $\mathcal{S}$, training with additional negative slices introduces gradients unrelated to accurate mask boundary detection, negatively impacting segmentation performance. To mitigate this issue, we implemented the gradient cut-off design (see Fig.2), preventing irrelevant negative gradients from propagating back into the segmentation module, thus ensuring segmentation quality remains consistent despite the presence of numerous negative slices. The classification query vector $q$ is updated in each layer as

$$q_{l+1} = \text{FFN}(\text{LN}(q_l + \text{CrossAttn}(q_l, sg([S_l, T_l])))), \tag{4}$$

where LN is layer normalization, $S_l$ and $T_l$ are segmentation queries and text embeddings after the $l$-th layer, and $sg$ is the stop-gradient operator. We show

in Sec. 4.3 that the gradient cut-off strategy significantly improved segmentation performance.

### 3.5   Inference on 3D Volumes

During training, all 3D volumes were processed into 2D images with FVE. Images from different volumes were mixed together in batches, and we randomly sampled a fixed number of prompts for each image per training iteration. During inference time, we leveraged the information shared in a 3D volume to achieve efficient inference. Given a 3D image and a list of prompts, we first computed the text embeddings for the prompts. The same embeddings were expanded for all 2D FVE-encoded images in the volume. We also repeat-interleaved the multiscale visual features and the mask features from each slice in the batch dimension to match the number of prompts. The image features and text embeddings for all slice-prompt pairs are then fed into the segmentation and classification module (upper right box in Fig. 2).

After the model outputs the segmentation masks and existence probabilities for each slice, we simply stacked the 2D masks together into the 3D volume. When multiple prompts for disjoint objects were presented, we first applied slice-level non-maximum suppression to remove overlapping objects and assigned each voxel to the class with the highest probability. We did not perform any further smoothing or processing in this work.

In order to provide options for better instance splitting, *BiomedParse-V* was able to output the edge of the prompted object in addition to the full segmentation mask. We used part of the segmentation queries to predict the edge while keeping the others for the original segmentation task. The same set of losses was used for the two targets during training. During instance segmentation inference, the user could optionally remove the edge from the predicted masks so that adjacent instances were separable. The masks expanded to their original area after being split into single-connected components. All operations were performed in stacked 3D volume. Due to the trade-off in removing small objects, we did not incorporate edge removal in our main experiments.

## 4   Preliminary Studies

To investigate the 3D text-guided segmentation capability of *BiomedParse-V* before training at scale, we conducted preliminary experiments on abdominal CT and MRI data with multiple anatomies. We focused our training and evaluation for cross-modality experiments on AMOS22 [8] and compared to supervised models and universal medical image segmentation models. We also performed ablation studies to understand the effectiveness of each component in *BiomedParse-V*.

### 4.1   Setup

We processed all 3D volumes into RGB images with the proposed fractal encoding technique. Each RGB image corresponds to one encoded slice in the volume,

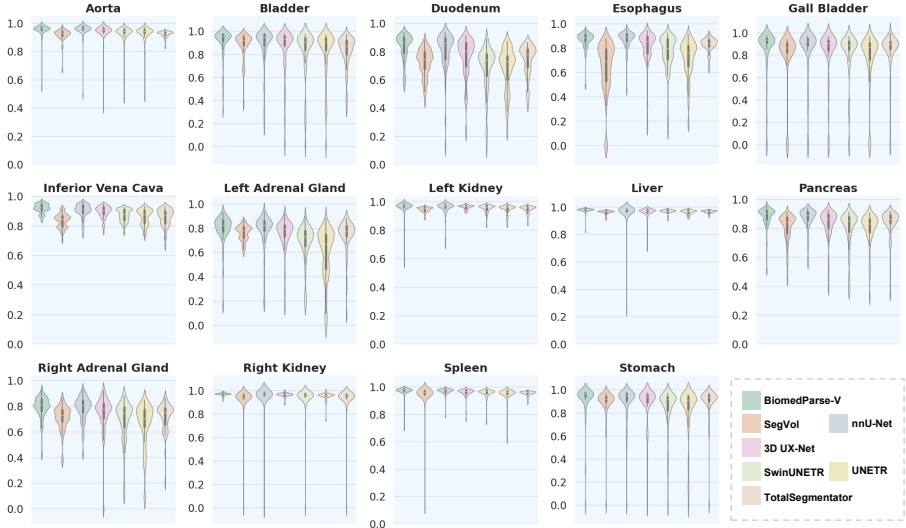

**Fig. 5.** Violin plots of Dice score evaluation for anatomies in CT, with *BiomedParse-V* compared with baseline models.

and we perform segmentation on the image. In this experiment we used default resolution of $1024 \times 1024$ to cover the highest resolution in AMOS22. We pair encoded images with text prompts following BiomedParse [28], and used the same data splits. Example prompts include "esophagus in abdominal computed tomography" and "stomach in abdominal magnetic resonance imaging". An empty mask prediction is expected where the prompted object is absent.

In terms of model architecture, we followed BiomedParse [28] and used FocalNet [25] as the image backbone. We adapted BoltzFormer [29] as the segmentation decoder head, adding the ISD module for object existence classification. The language encoder is a lightweight UniCL [26] architecture trained together with all other parts of the model. We initialized the image backbone from the pretrained SEEM [32] model, and all other parts trained from scratch.

### 4.2    Baseline comparisons

We compared *BiomedParse-V* with two types of baseline 3D segmentation models: (1) pretrained 3D segmentation foundation models, including SegVol[3] and TotalSegmentator [24]; and (2) supervised task-specific models, including nnU-Net [7], 3D UX-Net [11], SwinUNETR [4], and UNETR[5]. Table 1 presents the Dice scores for 14 anatomical structures in the CT dataset, as well as the average Dice scores for *BiomedParse-V* and all baseline models. Fig. 5 shows the violin plots of the Dice score distribution for the models on all anatomies. We present Dice scores on the MRI dataset in Table 2.

**Table 1.** Quantitative results of CT segmentation performance in Dice score (%) across *BiomedParse-V* and state-of-the-art models.

| Target | *BiomedParse-V* | SegVol | nnU-Net | 3DUX-Net | SwinUNETR | UNETR | TotalSeg. |
|---|---|---|---|---|---|---|---|
| Aorta | **95.27** | 92.07 | 95.20 | 94.00 | 92.85 | 92.84 | 92.53 |
| Bladder | **90.17** | 88.03 | 87.52 | 84.85 | 82.39 | 81.74 | 82.82 |
| Duodenum | **83.27** | 72.49 | 80.72 | 76.46 | 68.13 | 67.85 | 74.10 |
| Esophagus | 87.11 | 64.47 | **87.31** | 81.58 | 76.27 | 72.07 | 82.68 |
| Gallbladder | **85.96** | 79.05 | 83.06 | 81.59 | 81.28 | 76.98 | 83.85 |
| Left A.G. | **79.48** | 76.31 | 78.06 | 75.26 | 68.21 | 58.64 | 74.03 |
| Left Kid. | **96.39** | 94.58 | 95.39 | 96.30 | 95.20 | 94.64 | 94.88 |
| Liver | **97.71** | 96.24 | 96.09 | 95.83 | 96.89 | 96.50 | 96.70 |
| Pancreas | **88.42** | 80.97 | 86.57 | 82.53 | 79.42 | 79.60 | 84.29 |
| IVC | **92.02** | 83.65 | 90.38 | 89.56 | 87.30 | 85.99 | 85.79 |
| Right A.G. | **79.39** | 71.07 | 78.24 | 75.06 | 69.33 | 65.86 | 71.70 |
| Right Kid. | **96.88** | 92.92 | 93.19 | 96.38 | 93.46 | 95.30 | 93.22 |
| Spleen | **96.91** | 94.03 | **96.91** | 95.83 | 95.73 | 95.02 | 95.54 |
| Stomach | **91.49** | 88.82 | 89.79 | 86.96 | 84.32 | 82.34 | 89.68 |
| **Average** | **90.03** | 83.91 | 88.35 | 86.59 | 83.63 | 81.81 | 85.84 |

**Pretrained 3D segmentation foundation models.** We evaluated the performance of large-scale pretrained models SegVol and TotalSegmentator. SegVol originally trained exclusively on CT data and utilized here in text-prompt mode, achieved an average Dice score of 83.91. TotalSegmentator, benefiting from multi-class pretraining on the AMOS22 dataset, achieved an average Dice score of 85.84 on the CT dataset and 76.43 on the MRI dataset. Although both models leverage the advantages of extensive pretraining to generate robust generic representations, the inference performance demonstrates the gap between pretraining tasks and task-specific segmentation. This comparison underscores that while large-scale pretraining provides a strong starting point, tailoring the network architecture and task-specific training regimen is crucial to achieve state-of-the-art performance.

**Supervised task-specific models.** In addition to foundation models, we benchmarked *BiomedParse-V* against state-of-the-art supervised architectures, including nnU-Net, 3D UX-Net, SwinUNETR, and UNETR. For CT data, nnU-Net and 3D UX-Net achieved average Dice scores of 88.35 and 86.59, respectively, with SwinUNETR and UNETR slightly behind. *BiomedParse-V* surpassed the performance of all supervised approaches by achieving the highest average Dice score of 90.03 in all anatomical structures. In the MRI dataset, *BiomedParse-V* maintained its leading performance with an average Dice score of 84.94, outperforming its supervised counterparts by a significant margin.

**Quantitative Evaluation across Anatomies.** A detailed examination of the segmentation performance across varied anatomical structures reveals distinct performance gradients among the seven methods. The violin plots in Fig. 5 indicate that *BiomedParse-V* and nnU-Net exhibit higher median Dice scores

**Table 2.** Quantitative results of MRI segmentation performance in Dice score (%) across *BiomedParse-V* and state-of-the-arts.

| Target | *BiomedParse-V* | nnU-Net | 3DUX-Net | SwinUNETR | UNETR | TotalSeg. |
|---|---|---|---|---|---|---|
| Aorta | 95.73 | 95.64 | **96.23** | 96.13 | 94.66 | 85.71 |
| Duodenum | **76.03** | 66.78 | 69.58 | 66.40 | 62.44 | 55.50 |
| Esophagus | 81.38 | 73.62 | **82.76** | 81.40 | 74.12 | 79.26 |
| Gallbladder | **66.58** | 66.32 | 60.29 | 64.87 | 55.23 | 60.62 |
| Left A.G. | 63.35 | 57.15 | **68.88** | 64.90 | 62.11 | 57.79 |
| Left Kid. | **96.92** | 95.82 | 96.58 | 96.11 | 93.04 | 91.11 |
| Liver | 97.66 | 97.25 | **97.84** | 97.79 | 96.71 | 91.89 |
| Pancreas | **88.70** | 79.29 | 83.68 | 85.35 | 81.47 | 76.02 |
| IVC | 87.26 | **90.66** | 87.76 | 86.19 | 82.88 | 67.42 |
| Right A.G. | 68.14 | 53.29 | **69.61** | 65.94 | 56.44 | 56.85 |
| Right Kid. | **96.69** | 85.48 | 96.48 | 96.09 | 95.57 | 92.97 |
| Spleen | 96.88 | 96.66 | 97.13 | **97.16** | 94.20 | 89.91 |
| Stomach | **88.93** | 88.80 | 86.87 | 88.91 | 83.63 | 88.48 |
| **Average** | **84.94** | 80.52 | 84.13 | 83.63 | 79.42 | 76.43 |

with tighter distributions, suggesting a more refined and consistent segmentation performance across different organs. In contrast, the foundation models, TotalSegmentator and SegVol, show lower median values and higher variability, reflecting less robust segmentation performance. 3D UX-Net and SwinUNETR, demonstrate a gradual improvement in their performance distributions. This quantitative evaluation confirms that *BiomedParse-V* not only achieves the highest overall Dice scores, but also delivers consistent performance across diverse anatomical regions.

### 4.3   Ablation studies

We conducted ablation studies on the components of *BiomedParse-V* to assess their impact on segmentation performance. Unless otherwise specified, we used the default setting with a fractal encoding order of $k = 1$ and all modules enabled. For ablation experiments requiring additional training, we downsampled images to $512 \times 512$ to generate lower-resolution masks, which were then interpolated to match the ground truth size. This resulted in a slight overall performance drop compared to the main results.

**Fractal encoding order**  To explore the effect of different orders of fractal encoding and validate the need for spatial context, we trained the model in settings that range from no encoding (single slice) to encoding order of up to 3. We report the averaged Dice score reported in Table 3. We can see that the segmentation performance dropped when there was no extra encoded spatial context. The performance change is small when the order of encoding is changed from 1 to 3. This could be explained by the trade-off of more context on the

$z$ axis versus less resolution on the $x$-$y$ plane. We also note that the advantage of the fractal encoding was greater on MRI than on CT, possibly due to the spacing between adjacent slices from the two different imaging techniques.

**Table 3.** Segmentation performance in Dice score (%), without and with FVE of different orders $k$.

| Modality | No FVE | $k = 1$ | $k = 2$ | $k = 3$ |
|---|---|---|---|---|
| CT | 88.24 | **88.94** | 88.65 | 88.61 |
| MRI | 82.49 | 83.80 | **83.92** | 83.22 |

**Independent segmentation discriminator** To demonstrate the necessity of the ISD module, we configured *BiomedParse-V* to produce raw 2D segmentation masks without applying the slice filtering process in Eq. (2). As shown in Table 4, removing the ISD module led to a substantial drop in segmentation performance, particularly for CT. The performance decrease for MRI was notably smaller, likely because MRI volumes generally contain fewer empty slices. In contrast, CT scans typically cover a larger anatomical region, making slice filtering more impactful.

**Table 4.** Volumetric segmentation performance in Dice score (%), with raw 2D mask prediction and with the ISD slice filtering.

| Modality | Raw mask | ISD (ours) |
|---|---|---|
| CT | 73.87 | **90.03** |
| MRI | 82.31 | **84.94** |

**Gradient cut-off** To validate the effectiveness of the gradient cut-off design in the ISD module, we trained and evaluated the model with free back-propagation, and compared it with our proposed approach in Table 5. The volumetric segmentation performance decreased significantly when there was no stop-gradient operation from the segmentation module to the classification module, indicating the necessity of the design.

## 5    Experiments

In our main training experiment, we trained *BiomedParse-V* at scale on five different 3D modalities, and compared with other text-guided segmentation models.

**Table 5.** Volumetric segmentation performance in Dice score (%), with and without gradient cut-off in the ISD module.

| Modality | No cut-off | With cut-off (ours) |
|----------|------------|---------------------|
| CT       | 86.40      | **88.94**           |
| MRI      | 80.68      | **83.80**           |

### 5.1   Dataset and evaluation metrics

The development set is an extension of the CVPR 2024 MedSAM on Laptop Challenge [17], including more 3D cases from public datasets[4] and covering commonly used 3D modalities, such as Computed Tomography (CT), Magnetic Resonance Imaging (MRI), Positron Emission Tomography (PET), Ultrasound, and Microscopy images. The hidden testing set is created by a community effort in which all the cases are unpublished. The annotations are provided by the data contributors or annotated by the challenge organizer with 3D Slicer [9] and MedSAM2 [18]. In addition to using all training cases, the challenge contains a coreset track, where participants can select 10% of the total training cases for model development.

The text-guided segmentation task contains both semantic segmentation and instance segmentation. For the semantic segmentation task, the evaluation metrics include the Dice Similarity Coefficient (DSC) and Normalized Surface Distance (NSD) to evaluate the segmentation region overlap and boundary distance, respectively. For the instance segmentation task, we computed the F1 score at an overlapping threshold of 0.5 and DSC scores for true positives. In addition, the algorithm runtime will be limited to 60 seconds per class. Exceeding this limit will cause all DSC and NSD metrics to be set to 0 for that test case.

### 5.2   Implementation details

**Preprocessing** Following the practice in MedSAM [16], all images were processed to npz format with an intensity range of $[0, 255]$. Specifically, for CT images, we normalized Hounsfield units using typical window width and level values: soft tissues (W:400, L:40), lung (W:1500, L:-160), brain (W:80, L:40), and bone (W:1800, L:400). Subsequently, the intensity values were rescaled to the range of $[0, 255]$. For other images, we clipped the intensity values in the range between the 0.5th and 99.5th percentiles before rescaling them to the range of $[0, 255]$. If the original intensity range is already in $[0, 255]$, no preprocessing was applied.

For the training data, we followed Sect. 3.2 to process all 3D volumes into 2D RGB images. Based on ablation study, Sect. 4.3, we chose the fractal order $k = 1$ to balance the encoding depth and neighboring details. The smaller fractal order also allows more aggressive data augmentation while satisfying the conditions for Theorem A, as the size of the encoded "super pixels" are smaller.

---

[4] A complete list is available at https://medsam-datasetlist.github.io/

We adaptively chose the optimal views for slicing. For 3D image of shape $H \times W \times D$ on $x - y - z$ coordinate, we select the axial view ($x - y$ plane) by default if the ratio $\frac{2|H-W|}{H+W} < 0.5$, otherwise we select the view with the lowest ratio. We incorporate additional views during training if the ratio is less than 0.5 for all views and if the voxel spacing differs less than 10% across the dimensions. We converted all sliced images and masks to $512 \times 512$ resolution in 8-bit RGB format for efficient data processing, storage, and loading.

**Environment settings**  The development environments and requirements are presented in Table 6.

**Table 6.** Development environments and requirements.

| | |
|---|---|
| System | Ubuntu 22.04 |
| CPU | AMD EPYC 7V12 |
| RAM | 900 GiB |
| GPU | 40 NVIDIA A100 40G |
| CUDA version | 11.8 |
| Programming language | Python 3.10 |
| Deep learning framework | torch 2.3.1, torchvision 0.18.1 |

**Training protocols**  We trained *BiomedParse-V* on 40 NVIDIA A100 GPUs with 40 GB memory per GPU. The effective batch size counts to 320. For each 2D image example, we sample 4 classes from the feasible list of the source dataset, and randomly select one prompt for each class. The masks corresponding to the sampled classes were used as the ground truth for prediction, with absent classes resulting in empty masks (all-zero). We duplicated the image features across the 4 prompts for efficient computation.

We used AdamW [14] as the optimizer with the same segmentation loss and slice level classification loss. The segmentation loss was equally weighted Dice loss and pixel-wise binary cross-entropy loss. The classification loss was a slice-level binary cross-entropy loss with a positive class weight equal to 3. We used learning rate $1 \times 10^{-5}$ and weight decay $10^{-2}$. We applied an additional segmentation loss with the same setting on the edge of the masks for instance edge prediction.

# 6  Results and discussion

We evaluated *BiomedParse-V* and compared with baseline text-guided 3D biomedical image segmentation models CAT [6] and SAT [30]. We compared with the baseline model trained on all data as well as the core set.

**Table 7.** Training protocols.

| | |
|---|---|
| Pre-trained Model | SEEM |
| Batch size | 320 |
| Patch size | 4×4×3 |
| Total epochs | 80 |
| Optimizer | AdamW |
| Initial learning rate (lr) | $1 \times 10^{-5}$ |
| Lr decay schedule | Cosine |
| Training time | 240 hours |
| Loss function | Dice, Pixel BCE, Slice BCE |
| Number of model parameters | 371M |
| Number of flops | 8612G per slice |

**Table 8.** Quantitative evaluation results of the validation set on the **all-data track**.

| Modality | Method | Sematic Segmentation | | Instance Segmentation | |
|---|---|---|---|---|---|
| | | DSC | NSD | F1 | DSC TP |
| CT | CAT | 0.7211 | 0.7227 | 0.2993 | 0.3717 |
| | SAT | 0.6780 | 0.6726 | 0.2517 | 0.3954 |
| | *BiomedParse-V* | **0.8512** | **0.8965** | **0.5119** | **0.6749** |
| MRI | CAT | 0.5415 | 0.6193 | 0.1375 | 0.2813 |
| | SAT | 0.5610 | 0.6669 | 0.1228 | 0.2728 |
| | *BiomedParse-V* | **0.7396** | **0.8664** | **0.5317** | **0.7053** |
| Microscopy | CAT | - | - | 0.0313 | 0.3628 |
| | SAT | - | - | **0.2006** | 0.4243 |
| | *BiomedParse-V* | - | - | 0.1939 | **0.6552** |
| PET | CAT | - | - | 0.1098 | 0.2779 |
| | SAT | - | - | **0.4200** | **0.7863** |
| | *BiomedParse-V* | - | - | 0.3132 | 0.7185 |
| Ultrasound | CAT | 0.8594 | 0.8360 | - | - |
| | SAT | 0.8558 | 0.7924 | - | - |
| | *BiomedParse-V* | **0.9050** | **0.9135** | - | - |

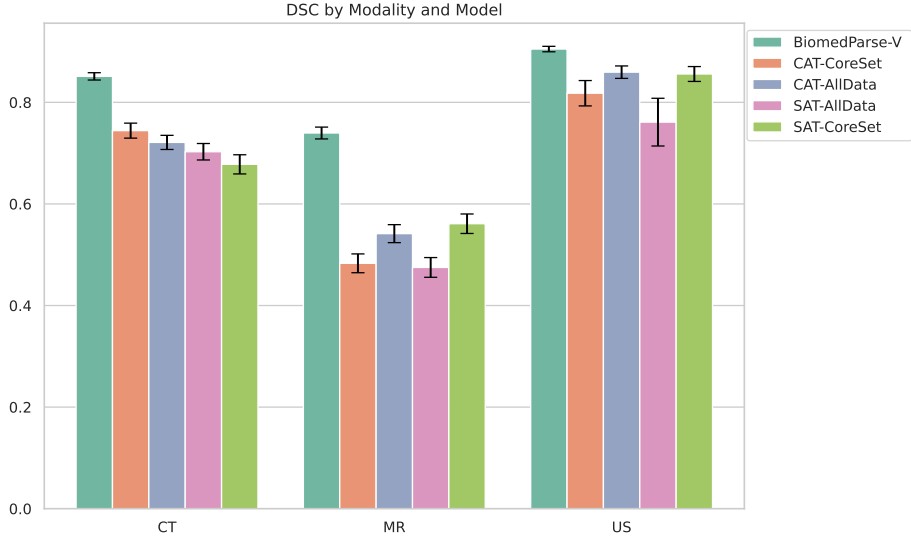

**Fig. 6.** Bar plot of semantic segmentation Dice Similarity Coefficients for *BiomedParse-V* and baseline models on different modalities.

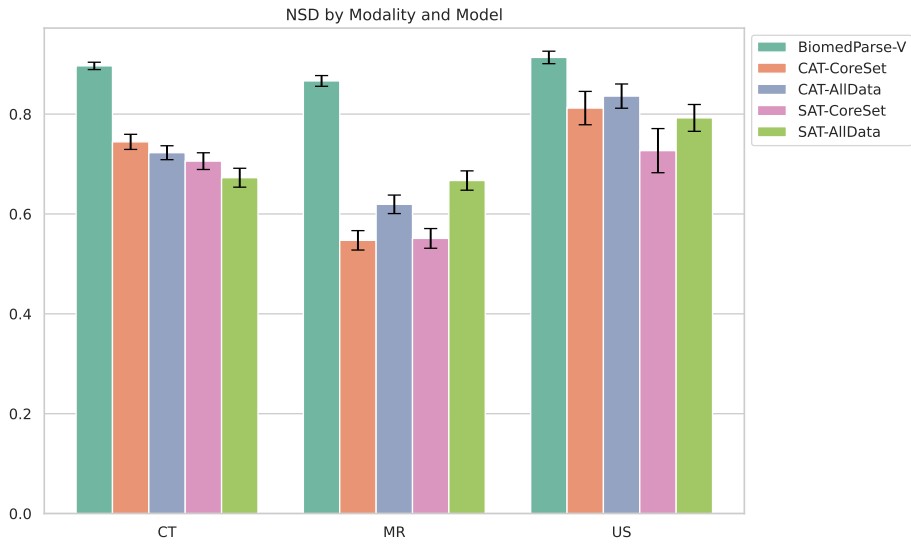

**Fig. 7.** Bar plot of semantic segmentation Normalized Surface Distances for *BiomedParse-V* and baseline models on different modalities.

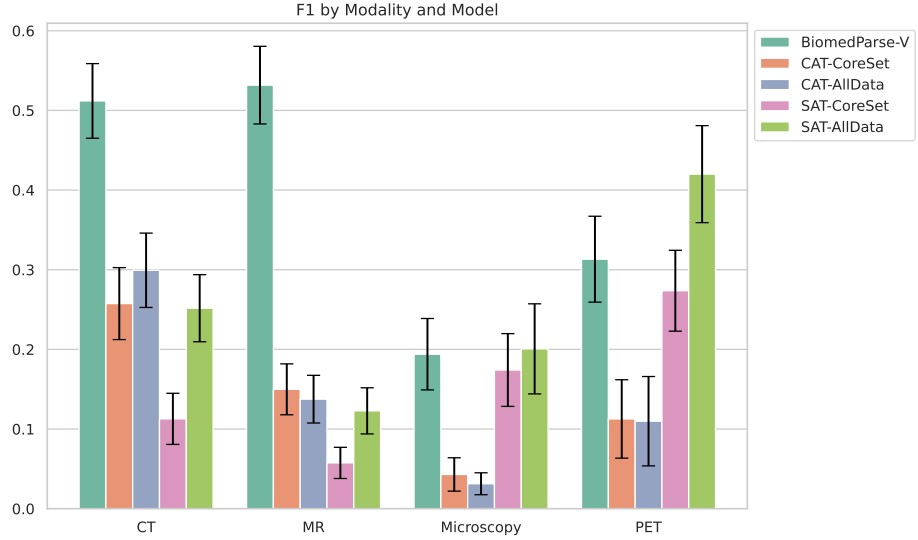

**Fig. 8.** Bar plot of instance segmentation F1 scores for *BiomedParse-V* and baseline models on different modalities.

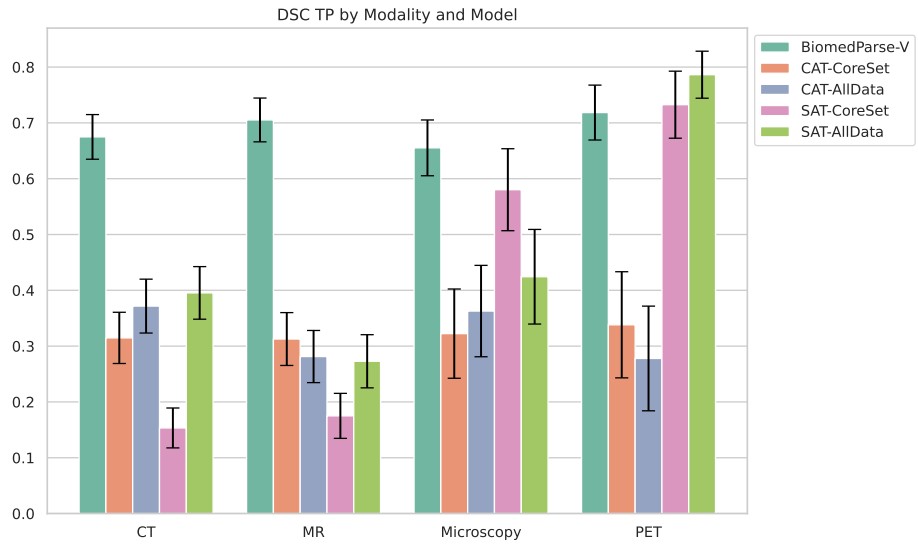

**Fig. 9.** Bar plot of instance segmentation Dice Similarity Coefficients on true positives for *BiomedParse-V* and baseline models on different modalities.

### 6.1 Quantitative results on validation set

As shown in Fig. 6, 7, 8, 9, and Table 8, *BiomedParse-V* outperformed baseline models in most modalities and metrics. The advantage is dominating for CT, MRI and ultrasound, where the margin ranges from 5-43%. For microscopy, *BiomedParse-V* outperforms all baseline models by 23-29% in terms of DSC on true positive instances, and on par with the best competing method with less than 1% difference in F1 score. *BiomedParse-V* outperforms CAT significantly, but lags behind SAT on PET with a 7-11% difference between metrics. In terms of computation, the average raw model inference time per volume on the validation set was 5.3 seconds, and 20.5 seconds including full environment and model loading.

### 6.2 Qualitative results on validation set

Representative visualizations are shown in Figs. 10, 11, 12, and 13, including two successful and two suboptimal cases. In the first two, *BiomedParse-V* correctly segments all prompted objects despite cluttered scenes or low image quality.

In Fig. 12, *BiomedParse-V* recovers most annotated nuclei but tends to undersegment low-intensity nuclei. Touching or overlapping nuclei are sometimes merged, which complicates downstream instance separation. In our experiments, edge-removal post-processing alleviated some merges but occasionally removed small true positives as a trade-off.

In Fig. 13, *BiomedParse-V* was prompted to segment the right humerus, yet the ISD module predicted absence in the slice. A likely failure mode is anatomical symmetry, which makes laterality ambiguous. Because *BiomedParse-V* operates on locally encoded 2D slices with limited 3D context, and training includes image-level augmentations (e.g., left–right flips), laterality cues can be weakened, leading to such false negatives.

### 6.3 Results on final testing set

In the hidden testing set, *BiomedParse-V* consistently outperformed all baseline methods across all metrics. As shown in Table 9, *BiomedParse-V* exhibits a 10-46% advantage. As an overall observation, instance segmentation is still a challenging task compared to semantic segmentation, as it involves more tumor and lesion segmentation cases, which are typically small in size and exhibit high variance in location.

### 6.4 Limitation and future work

While *BiomedParse-V* tackles the 3D segmentation problem by encoding 3D context into 2D images with FVE, the total amount of spatial information is upper bounded by a native 3D architecture. As a result, it still lags behind the best competing model when the full volumetric context is essential. We envision future

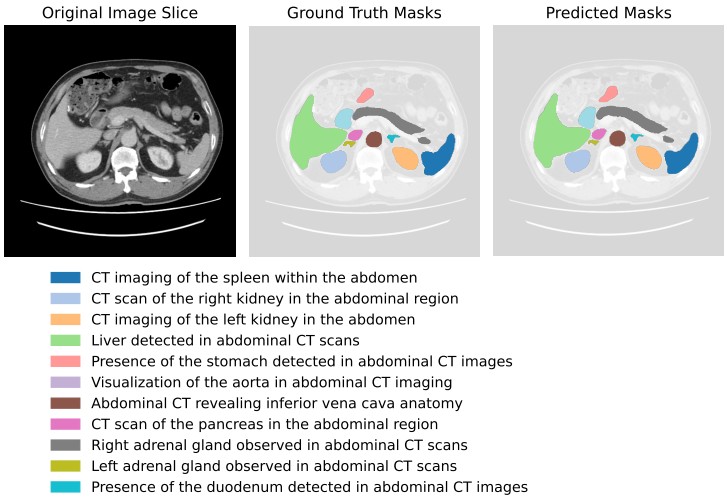

Fig. 10. Visualization of organ segmentation in abdominal CT.

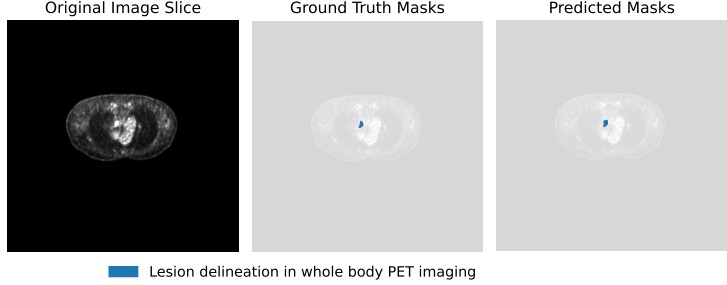

Fig. 11. Visualization of lesion segmentation in PET.

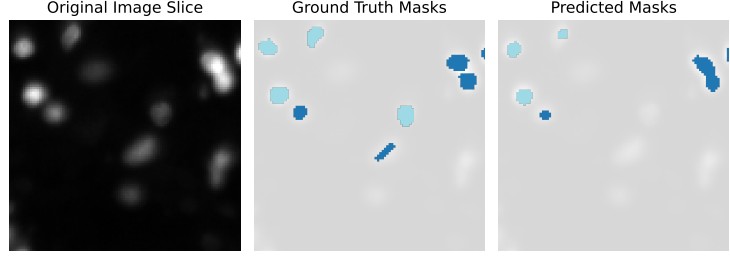

Fig. 12. Visualization of nuclei segmentation in microscopy.

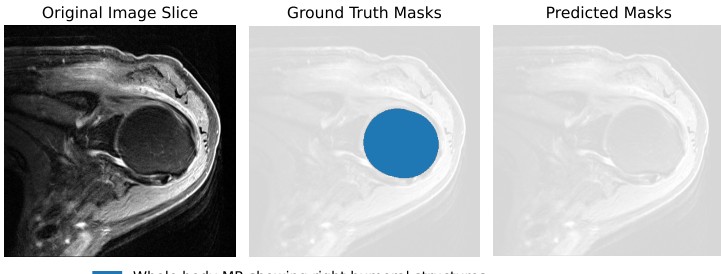

Fig. 13. Visualization of humerus segmentation in MRI.

Table 9. Quantitative evaluation results of the testing set on the **all-data track**.

| Method | Sematic Segmentation | | Instance Segmentation | |
|---|---|---|---|---|
| | DSC | NSD | F1 | DSC TP |
| CAT | 0.3304 | 0.3153 | 0.0194 | 0.0466 |
| SAT | 0.5413 | 0.5297 | 0.1419 | 0.2959 |
| *BiomedParse-V* | **0.7497** | **0.7747** | **0.2380** | **0.4476** |

models to be able to efficiently encode richer 3D context, or take the BoltzFormer segmentation framework to the native 3D domain.

On the other hand, since *BiomedParse-V* only outputs binary segmentation masks, the heuristic instance splitting approach limits its performance, as shown in the instance segmentation results. We envision the model architecture to be expanded to produce more direct and precise instance predictions, or to combine with existing instance segmentation tools to achieve best performance.

### 6.5 Discussions

In order to tackle the problem of 3D medical image segmentation driven by text prompting, we presented generic applicable approaches to utilize 2D foundation models for 3D images. With a rich spatial context encoded in a single image, vision models with downstream applications such as classification, text generation, and question answering could be directly applied to solve 3D problems.

In addition, our fractal-based encoding approach is not limited to 3D medical images but can extend to other vision domains that involve sequential or volumetric data, such as video-based analysis. Thus, Fractal Volumetric Encoding (FVE) provides a versatile and efficient solution, unleashing the power of vision foundation models across various application domains.

The Independent Segmentation Discriminator (ISD) module with gradient cut-off was proven to be effective in determining object existence while maintaining coherence segmentation performance. The same idea could be applied to applications such as video segmentation, where object existence is also a challenge [20].

## 7   Conclusion

We presented *BiomedParse-V*, a novel and efficient multimodal model for 3D medical image segmentation that leverages the strengths of pretrained 2D foundation models. By employing Fractal Volumetric Encoding (FVE), our method compresses 3D volumes into compact 2.5D representations that preserve essential spatial context and readily fit common vision foundation models. Complemented by an Independent Segmentation Discriminator (ISD) with gradient cut-off, *BiomedParse-V* achieves enhanced volumetric segmentation accuracy. Extensive experiments on CT and MRI datasets demonstrate that our approach outperforms current state-of-the-art pretrained models and supervised expert models. We envision our work opening a promising direction for practically deploying robust foundation models in clinical medical imaging, addressing critical scalability and resource-constraint challenges inherent to volumetric segmentation.

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

# A   Proof of theorem

**Theorem 1 (Equivalence to 3D Convolution).** *Given a fractal encoding function $\mathcal{E}$, defined as*

$$\mathcal{E}(\mathcal{V}, i) : \mathbb{R}^{H \times W \times D} \otimes \mathbb{N} \to \mathbb{R}^{H \times W \times C}, \qquad (5)$$

*applying a linear embedding operator on patches of the encoded image corresponds mathematically to performing an adaptive-resolution 3D convolution on the original volume.*

**Proof of Theorem A**

*Proof.* We will show that for any spatial patch of the fractal-encoded image, the linear projection on that patch reproduces exactly an adaptive-resolution 3D convolution on the original volume.

*1. Notation.*

– Volume $\mathcal{V} \in^{H \times W \times D}$, focal index $i$.
– Encoding order $k$ induces patch size $p = 2^{k-1}$ and depth-extent $\Delta_k = 3k - 2$.
– Encoded image $\mathcal{I} = \mathcal{E}(\mathcal{V}, i) \in^{H \times W \times C}$, $C = 3$.
– Fix a 2D patch

$$P = \big\{ \mathcal{I}_{i_0+u, \, j_0+v, \, c} \ \big| \ 0 \le u, v < p, \ 0 \le c < C \big\}.$$

– Flattening $\mathrm{vec}(P) \in^{p^2 C}$; linear weights $W \in^{M \times (p^2 C)}$; embedding $e = W \,\mathrm{vec}(P) \in^{M}$.

*2. One-to-one mapping to voxels.* By the fractal-encode/pixel_mix construction, each element of $\mathrm{vec}(P)$ is exactly one voxel

$$\mathcal{V}_{x,y,z} \quad \text{with} \quad (x,y) \in \{i_0, \ldots, i_0+p-1\} \times \{j_0, \ldots, j_0+p-1\}, \ \ z \in \{i-\Delta_k, \ldots, i+\Delta_k\}.$$

Thus

$$\mathrm{vec}(P) = \big[\mathcal{V}_{x_\ell, \, y_\ell, \, z_\ell}\big]_{\ell=1}^{p^2 C} \quad \text{for a bijection } \ell \mapsto (x_\ell, y_\ell, z_\ell).$$

*3. Standard 3D convolution.* An adaptive-resolution 3D convolution at spatial location $(i_0, j_0, i)$ with kernel $K \in^{p \times p \times (2\Delta_k + 1)}$ computes

$$(K * \mathcal{V})_{i_0, j_0, i} = \sum_{u=0}^{p-1} \sum_{v=0}^{p-1} \sum_{d=-\Delta_k}^{\Delta_k} K_{u,v,d+\Delta_k} \, \mathcal{V}_{i_0+u, \, j_0+v, \, i+d}.$$

*4. Linear embedding matches convolution.* The patch embedding is

$$e = W \,\mathrm{vec}(P) = \sum_{\ell=1}^{p^2 C} W_{:,\ell} \, \mathcal{V}_{x_\ell, y_\ell, z_\ell} = \sum_{u,v,d} W_{(u,v,d)} \, \mathcal{V}_{i_0+u, \, j_0+v, \, i+d}.$$

By choosing

$$K_{u,v,d+\Delta_k} \ = \ W_{(u,v,d)}, \quad \forall\, u, v, d,$$

the two sums coincide, hence $e = (K * \mathcal{V})_{i_0, j_0, i}$.

*5. Adaptive-resolution property.* In fractal encoding, slices farther from $i$ are downsampled by factors of two per level—exactly mirroring a convolution whose spatial footprint shrinks for distant depth channels.

Therefore, every linear patch embedding on $\mathcal{E}(\mathcal{V}, i)$ reproduces an adaptive-resolution 3D convolution on $\mathcal{V}$.

## B   Fractal encoding examples

We visualize the encoded RGB images from FVE in Fig. 14 and 15. An encoding-decoding example is shown in Fig. 16. We show the voxel allocation with different orders of fractal encoding in Fig. 17. The majority portion of the voxels is from the slices close to the center slice. As the order increases, the size of a "super pixel" also increases. The ratio of the voxels from the close layers keeps the same when increasing the encoding order.

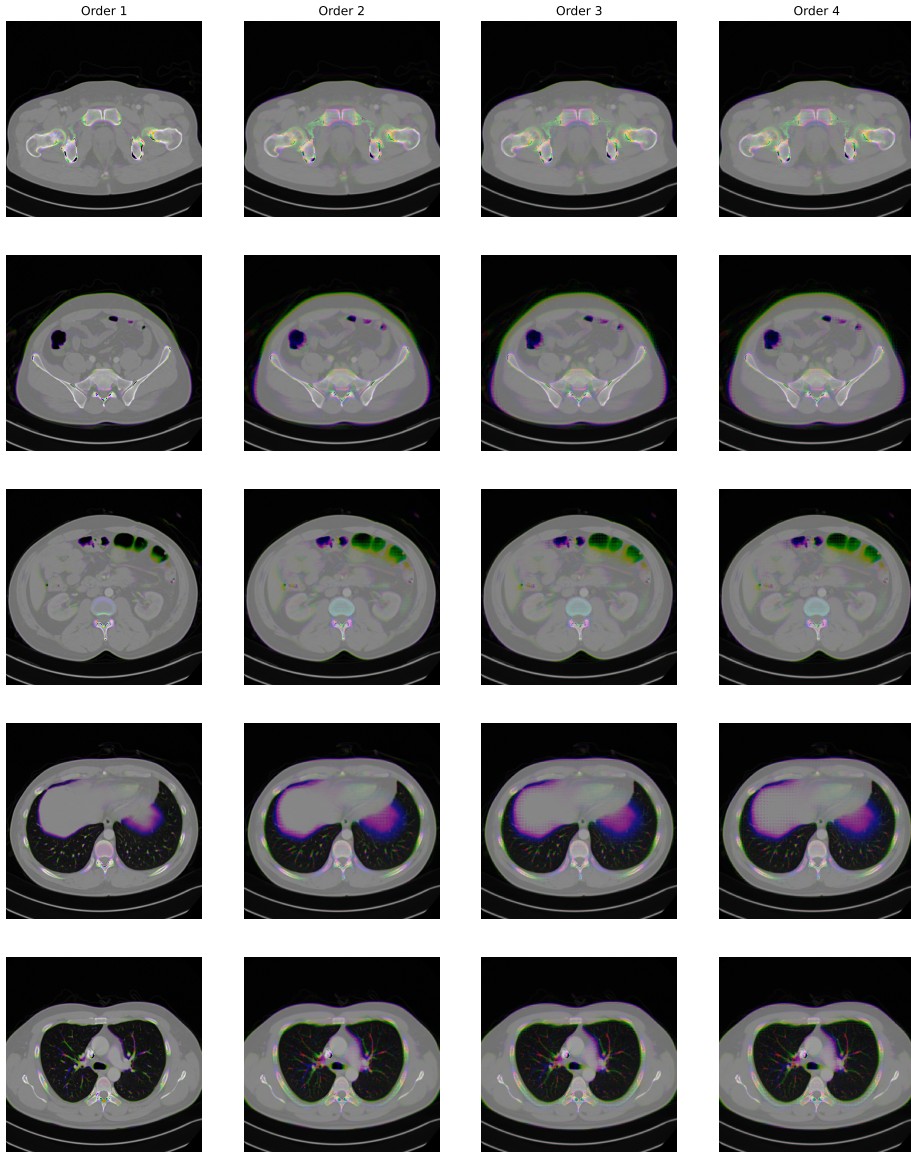

**Fig. 14.** Fractal encoded RGB image for CT with different encoding order. Each row corresponds to encoding around the same focal slice.

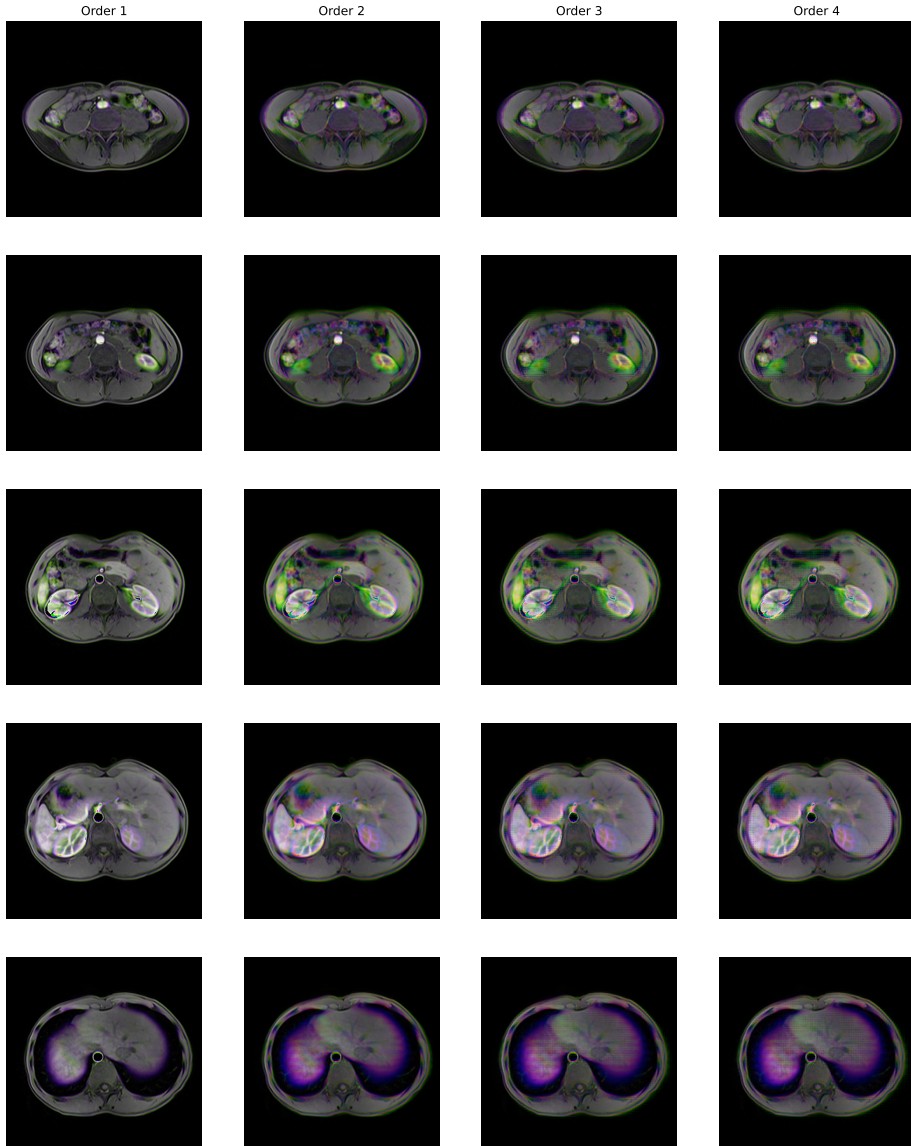

**Fig. 15.** Fractal encoded RGB image for MRI with different encoding order. Each row corresponds to encoding around the same focal slice.

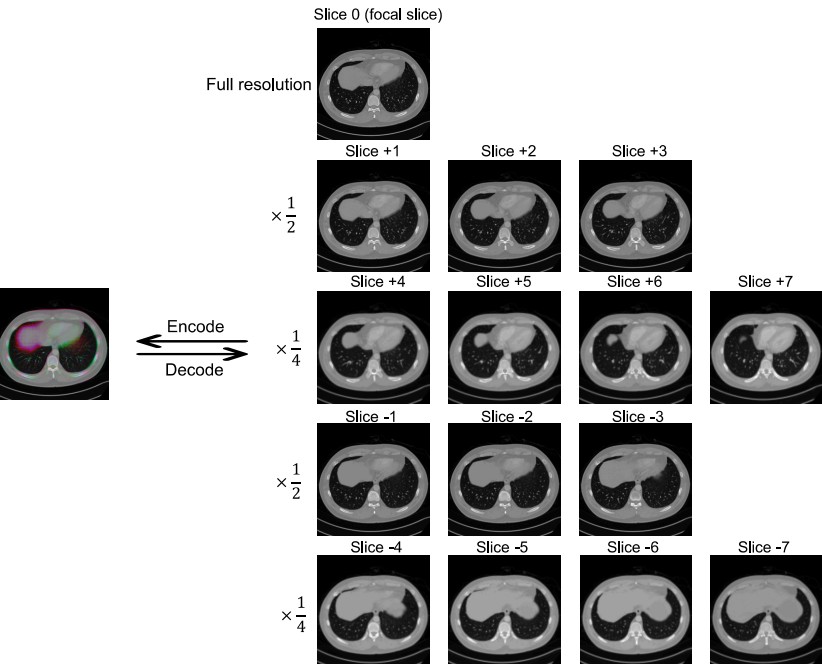

**Fig. 16.** Example encoding-decoding around a focal slice. The FVE encoded image on the left contains a full resolution focal slice along with adaptively downsampled context slices. The encoded slice can decode back to the slices on the right losslessly.

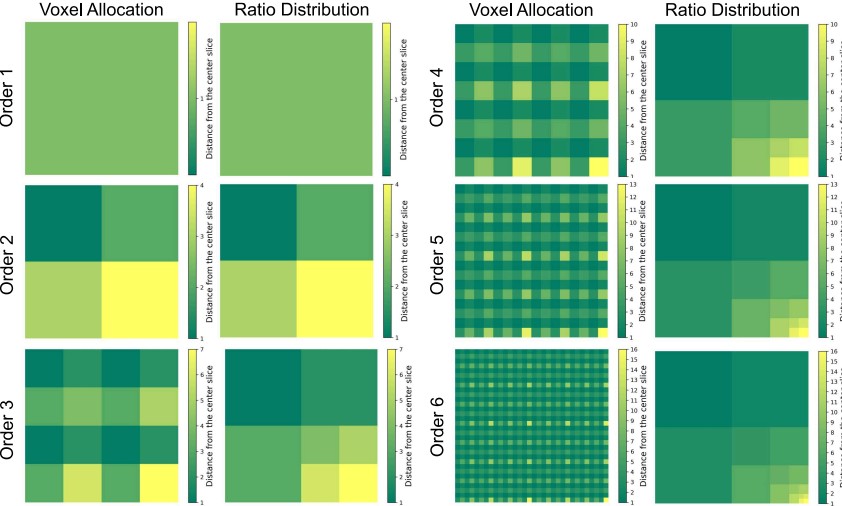

**Fig. 17.** The voxel allocation pattern for FVE of order 1 through 6. We show the minimal unit of neighborhood encoding (super pixel) for each order. The color represents the distance from the voxel to the focal slice in the original volume. The right side of each column shows the overall ratio distribution of the encoded voxels in terms of their distances to the focal slice.