# OpenReview forum: "BiomedParse-V : Scaling Foundation Model for Universal Text-guided Volumetric Biomedical Image Segmentation"
_thecvf.com/CVPR/2025/Workshop/MedSegFM — CVPR 2025 Workshop MedSegFM Submission_

### Official Review · Reviewer_w5Ew · 2025-09-13
**Review for BiomedParse-V**

**Rating:** 9
**Confidence:** 4

**Review:**

*Overview*

The paper proposes BiomedParse-V, a text-guided volumetric image segmentation model built on the pretrained 2D foundation model BiomedPars. The approach includes a novel Fractal Volumetric Encoding (FVE) module and an Independent Segmentation Discriminator module.

*Strengths*

The proposed Fractal Volumetric Encoding is interesting: it compresses and projects 3D images into 2.5D, enabling direct use of 2D foundation models.
According to Table 8, the method substantially outperforms CAT and SAT on the all-data track.

*Weaknesses*

1. From Table 3, performance appears to degrade as k increases (i.e., as more adjacent slices are considered). For example, k=1 (total of 7 slices) yields the best CT performance. This seems at odds with the common understanding that 3D spatial features and long-range context should help. Could this be related to the pixel-level mapping strategy? The current method directly selects values from four slices, potentially causing discontinuities across those sampled pixels.

2. A question about training details: batch size = 320 and patch size = 4×4×3—these refer to the image size? During training, is the image branch trained jointly, or handled separately?

3. Minor comments:
Figures 6–9 could be presented in a more polished, visually appealing style.
Page 18: “Representative visualizations are shown in Figs. 14, 11, 12, and 13, including” — should the first figure reference be Fig. 10?

---

> ### Author Rebuttal · Authors · 2025-11-05
>
> We sincerely thank the reviewer for the very positive and detailed feedback. We appreciate the recognition of our contributions, particularly the novelty of the Fractal Volumetric Encoding (FVE) and the Independent Segmentation Discriminator (ISD). Below are our responses to the specific comments.
>
> 1. Degradation with increasing fractal order k: We appreciate this insightful observation. Indeed, performance variation with larger k stems from the pixel-level mapping strategy used in FVE. As the total number of pixels packed in a single RGB image is fixed, the fractal order k serves as the trade-off parameter for encoding depth and resolution. As k goes higher, more slices are included, but the resolution of some encoded slices will decrease. Therefore, there exist a sweetspot where the trade-off is balanced. We observed that MRI, compared to CT, favors slightly deeper encoding, as the gap between two slices is typically smaller. We have clarified this explanation in the manuscript and discussed the tradeoff in details.
>
> 2. Clarification on training details: We thank the reviewer for requesting clarification. The batch size of 320 refers to the number of training samples per step. The patch size corresponds to the unit processed by the image backbone, which is FocalNet in this experiment. Each patch contains 4x4 pixels with 3 channels. The image and language branches are trained jointly in an end-to-end fashion, sharing gradients through the segmentation decoder. The details on the model architecture have been added to the revised experiment section for clarity.
>
> 3. We have revised the reference on the figures, and conducted a final proofread pass to ensure formatting accuracy throughout the manuscript.
>
> We thank the reviewer again for the encouraging evaluation and thoughtful feedback, which helped us strengthen both clarity and presentation.

---

### Official Review · Reviewer_AABu · 2025-09-17
**Review for BiomedParse-V**

**Rating:** 7
**Confidence:** 4

**Review:**

The paper introduces BiomedParse-V, a text-guided 3D biomedical image segmentation framework that leverages pretrained 2D vision models via Fractal Volumetric Encoding (FVE) and augments slice-wise consistency with an Independent Segmentation Discriminator (ISD). The method reports strong results across multiple modalities and tasks, notably surpassing CAT and SAT on the all-data track, and provides preliminary theoretical justification by relating FVE to adaptive 3D convolution.

## Strengths
1. Clever FVE design that packs volumetric context into a compact 2.5D representation, enabling reuse of mature 2D backbones at scale.
2. Clear problem motivation and practical training/inference pipeline; code-level details improve reproducibility.

## Weaknesses
1. Instance segmentation remains a weak point: performance relies on heuristic edge removal and post hoc splitting, with acknowledged trade-offs for small objects.
2. It seems that the preprocessing is modality-specific?
3 The efficiency claims could be better substantiated which is important in  this challenge.

---

> ### Author Rebuttal · Authors · 2025-11-05
>
> We sincerely thank the reviewer for the positive assessment and constructive feedback. We are glad that the reviewer found the proposed FVE design, motivation, and implementation clarity valuable. Below are our responses to the specific concerns.
>
> 1. Instance segmentation performance: We agree that instance segmentation remains a challenging aspect for complex biomedical volumes. In this work, we focused on unifying semantic and instance segmentation under a text-prompted framework, and the architecture was not specifically designed for instance segmentation. We provided options to remove edges for better instance splitting quality, anticipating to optimize for different settings. We've made the discussion in the limitation section.
>
> 2. Modality-specific preprocessing: We clarify that our preprocessing pipeline is largely shared across modalities, involving only normalization and intensity rescaling steps appropriate to each imaging type (CT, MRI, Ultrasound, etc.). The FVE module itself is modality-agnostic and directly operates on the 3D voxel tensors, without requiring modality-specific architecture or hyperparameter tuning.
>
> 3. Efficiency evaluation: We appreciate the comment on substantiating efficiency claims. In the revised manuscript, we now provide statistics for inference speed, which showed that the per-volume inference time (20s) is well below the threshold in the challenge.
>
> We thank the reviewer again for the insightful feedback and are encouraged by the positive evaluation of our work.

---

### Official Review · Reviewer_Phrt · 2025-09-20
**Review for BiomedParse-V**

**Rating:** 6
**Confidence:** 5

**Review:**

This paper presents BiomedParse-V, a prompt-driven foundation model for 3D biomedical image segmentation. Authors proposed a novel Fractal Volumetric Encoding (FVE) module to hierarchically compress 3D volumes into compact 2.5D RGB representations and another proposed Independent Segmentation Discriminator (ISD) for verifing slice-level object presence to reduce false positives. The framework supports text-guided semantic and instance segmentation across multiple imaging modalities (CT, MRI, PET, Ultrasound, Microscopy). Extensive experiments show BiomedParse-V achieves state-of-the-art Dice and NSD scores.

Strenth:
  1. The proposed FVE module tackles the challenge of high-dimensional 3D segmentation, combining with the ISD module, the whole framework enables a lower computational cost than conventional baseline models.
  2. The proposed framework enables text-guided segmentation and shows better performance than other related baseline models.

Weakness:
  1. Lack of text-guided ablation: Although the paper emphasizes text-guided segmentation, no ablation studies are provided to quantify the contribution of text prompts. Also, authors did not explicitly give the text prompts they used in experiments and did not discuss the importance of different prompts.
  2. Efficiency evaluation is incomplete: While this work is motivated as a more efficient 3D segmentation foundation model, the paper does not include concrete comparisons of parameter counts, inference speed, or memory usage against other 3D baselines.
  3. Confused backbone specification: The authors claimed to reuse large-scale 2D vision-language foundation models, but the paper does not explicitly provide which backbones are adopted (e.g., CLIP variants or SAM/MedSAM). Both text and figures lack clarity, which may hinder reproducibility and make it difficult to judge how much the results depend on the chosen pretrained model.

---

> ### Author Rebuttal · Authors · 2025-11-05
>
> We sincerely appreciate the reviewer’s valuable comments and constructive suggestions. We have carefully addressed each point and revised the manuscript accordingly. Below is our point-by-point response to the concerns raised.
>
> 1. As text-guided segmentation is the main setting of the challenge, the text prompt serves as the only form of instruction for the model to segment different objects. Since there are no predefined classes, the model always requires a text prompt describing the target object to perform segmentation. The prompts used in our experiments are concise natural-language descriptions of the object type present in the image (e.g., “left lung,” “liver lesion,” “femur bone”). We have clarified this in the revised manuscript.
>
> 2. We have now included quantitative statistics on inference speed and memory usage in the revised version. As BiomedParse-V was primarily designed for high-quality segmentation, efficiency was a secondary consideration; thus, the original manuscript emphasized accuracy. The newly added results provide a clearer view of runtime and computational efficiency for completeness.
>
> 3. We thank the reviewer for highlighting the need for more clarity on the model components. We have added a detailed paragraph describing the architecture and initialization scheme, along with a link to the released code for reproducibility:
>
> “In terms of model architecture, we followed BiomedParse and used FocalNet as the image backbone. We adapted BoltzFormer as the segmentation decoder head, incorporating the ISD module for slice-level object existence classification. The language encoder is a lightweight UniCL architecture trained jointly with the image backbone and decoder. The image backbone was initialized from the pretrained SEEM model, while all other modules were trained from scratch.”
>
> This clarification and code release should make it easier for the community to reproduce and build upon our results.
>
> We thank the reviewer again for the insightful feedback, which has helped us strengthen the paper in clarity, completeness, and reproducibility.

---

### Decision · Program_Chairs · 2025-11-12

Accept